# Anoxic Treatment of Agricultural Drainage Water in a Venturi-Integrated Membrane Bioreactor

**DOI:** 10.3390/membranes13070666

**Published:** 2023-07-14

**Authors:** Necati Kayaalp

**Affiliations:** Civil Engineering Department, Dicle University, 21280 Diyarbakir, Turkey; necati.kayaalp@dicle.edu.tr

**Keywords:** anoxic denitrification, membrane bioreactor (MBR), fouling, venturi, hydrogen

## Abstract

Due to low sludge production and being a clean source without residuals, hydrogen-based autotrophic denitrification appears to be a promising choice for nitrate removal from agricultural drainage waters or water/wastewater with a similar composition. Although the incorporation of hydrogen-based autotrophic denitrification with membrane bioreactors (MBRs) enabled almost 100% utilization of hydrogen, the technology still needs to be improved to better utilize its advantages. This study investigated the anoxic treatment of both synthetic and real drainage waters using hydrogen gas in a recently developed membrane bioreactor configuration, a venturi-integrated submerged membrane bioreactor, for the first time. The study examined the effects of the inflow nitrate concentration, and the use of a venturi device on the removal efficiency, as well as the effects of the presence of headspace gas circulation and circulation rate on membrane fouling. The study found that using the headspace gas circulation through a venturi device did not significantly affect the treatment efficiency, and in both cases, a removal efficiency of over 90% was achieved. When the inlet $NO_{3}^{-}–N$ concentration was increased from 50 mg/L to 100 mg/L, the maximum removal efficiency decreased from 98% to 92%. It was observed that the most significant effect of the headspace gas circulation was on the membrane fouling. When the headspace gas was not circulated, the average membrane chemical washing period was 5 days. However, with headspace gas circulation, the membrane washing period increased to an average of 12 days. The study found that the headspace gas circulation method significantly affected membrane fouling. When the upper phase was circulated with a peristaltic pump instead of a venturi device, the membrane washing period decreased to one day. The study calculated the maximum hydrogen utilization efficiency to be approximately 96%.

## 1. Introduction

### 1.1. Environmental Effects of Agricultural Drainage Waters

In agricultural areas where natural drainage is insufficient, the wastewater formed due to the removal of unused irrigation water from the land with drainage channels is called agricultural drainage water [1]. Agricultural drainage waters containing salt, trace elements and other pollutants can damage both soil and aquatic ecosystems when discharged into receiving water environments. Excessive nitrogen that is not used by plants and animals in agricultural systems can percolate into shallow groundwater and eventually enter surface waters through concentrated or diffuse discharges [2]. For example, 25% of 248 groundwater well samples taken from a large area in Mexico, which meets 10% of citrus production, exceeded the national drinking water standard of 10 mg/L $NO_{3}^{-}–N$ concentration [3].

Although many studies have been conducted on the reuse of domestic and industrial wastewater, little attention has been paid to the reuse of agricultural drainage waters after proper treatment. Drainage waters are reused in various parts of the world without treatment, but this may cause problems such as soil salinization, decreased permeability, rise in groundwater level, etc.

Since approximately 3/4 of the clean water used contributes to agricultural irrigation, these waters have great recovery potential. This rate is 70% worldwide [4], and 77% in Turkey [5]. For instance, in the measurements made by the 15th Regional Directorate of State Water Works (DSI), the amount of water removed by the main drainage channel in the Harran Plain (Sanliurfa, Turkey) has been reported to be 190–200 million m^3^/year in recent years [6]. This amount is considerably higher than the annual drinking water need of Sanliurfa, which had a population of 1,985,000 in 2017.

### 1.2. Treatment Methods for Agricultural Drainage Waters

Treatment of agricultural drainage waters is the last resort option in drainage water management, due to the high cost of treatment and the uncertainty of achieving the desired level of treatment [7]. One of the distinctive features of agricultural drainage waters is the insufficient carbon source required for biological treatment. Zhang et al. [8] investigated the ability of five different bacteria to use molasses as a carbon source in their study, in which they used molasses (sugar residue) as a carbon source to reduce selenate (Se^+6^) in the drainage water to elemental selenium (Se^0^) using bacteria. The study found that *Enterobacter taylorae* converts selenate to selenium with an efficiency of 97% in artificial drainage water containing 1000 μg/L selenate, and that molasses could be used as an economical carbon source in drainage water treatment.

Quinn et al. [9] established a demonstration plant with algal–bacterial treatment to remove nitrate and selenium from drainage water. The process achieved over 95% nitrate removal and 80% dissolved selenium removal. Hunt et al. [10] investigated the denitrification efficiency of biomass that adheres to polyvinyl alcohol (PVA) material in agricultural drainage water. The study reported a 50% removal efficiency during a 1 h soaking period in water containing 7.8 mg $NO_{3}^{-}–N$/L. The nitrate removal rate was also high, with a value of 94 g $NO_{3}^{-}–N$ /m^2^/day. 

Allred [11] investigated the removal of nitrate and phosphate from agricultural drainage waters through filtration using filter materials formed by reacting iron particles with zero-valent iron and sulfur. The intermittent experiments showed that both filter materials achieved at least 94% phosphate removal and 86–88% nitrate removal. Messer et al. [12] conducted a study utilizing two distinct wetland restoration sites with different soils (organic and mineral soils) for treating $NO_{3}^{-}–N$-rich synthetic agricultural drainage water. The results showed that the mineral soil wetland achieved a better performance than organic soil in terms of nitrate removal, which was attributed to higher plant uptake in the mineral soil due to its limited nitrogen content. In another study, Lavrnic et al. [12] examined the long-term performance of a full-scale surface flow-constructed wetland in terms of nitrogen removal rates, among other parameters. Two-year monitoring showed up to 78% removal of $NO_{3}^{-}–N$.

### 1.3. Membrane Bioreactors

According to Judd [13], all processes that combine a selective membrane with biological water and wastewater treatment to capture solids are called membrane bioreactors (MBRs). The use of MBRs in water and wastewater treatment has some advantages, such as high biomass concentrations and high organic loading rates due to high sludge ages, better output quality in terms of turbidity, bacteria, particulate and colloidal organic matter, and a reduction in treatment cost of industrial sludges due to lower sludge production [14,15]. However, membrane fouling and the associated high operating cost can be considered the main disadvantages of MBRs.

Membrane fouling is the clogging of the membrane pores due to foulants accumulating on the membrane, leading to a reduction in the membrane flux and an increase in transmembrane pressure. When the membrane fouling reaches an advanced level, the membrane cannot be used further, and it must be cleaned physically and chemically. Membrane fouling can have many reasons depending on the inlet wastewater characteristics, the membrane and module itself, and operating conditions.

The most widely used method to reduce membrane fouling is to scour the membrane surface with a gas [16]. This is valid for both aerobic and anaerobic membrane bioreactors [17]. While air is mostly used in aerobic reactors (e.g., [18]), headspace gas is generally used in anaerobic conditions [19].

### 1.4. Hydrogenotrophic Denitrification

In the removal of nitrates from agricultural drainage waters by heterotrophic denitrification, an external organic carbon source needs to be added. Several organic carbon sources such as methanol [10], acetate, lactate, and glucose [20] were used in previous studies. However, some of this externally added organic carbon is utilized by selenate-reducing bacteria, and most of it is removed with the effluent stream, causing an increase in the cost of treatment due to the wasted expensive chemicals [20]. As an alternative economic source of organic carbon, Zhang et al. [21] investigated rice straw, and Zhang et al. [8] studied the usability of molasses.

In agricultural drainage waters with an insufficient carbon content, nitrate removal using the autotrophic denitrification process, which does not require an organic carbon source, has significant potential. The autotrophic denitrification processes can be divided into two basic groups based on the electron donor: hydrogen-based and sulfur-based reactions. However, the requirement for limestone to produce sulfates and for pH adjustment severely limits the applicability of sulfur-based denitrification. Hydrogen-based denitrification by species such as *Ochrobactrum anthropi*, *Pseudomonas strutzeri*, *Paracoccus denitrificans* and *Paracoccus panthotrophus* has two main advantages over sulfur-using denitrification and heterotrophic denitrification. One advantage of hydrogen-based denitrification is that it is practically impossible for the given electron donor to be left behind, since when water (or wastewater) is exposed to the open surface, H_2_ is released into the atmosphere. Additionally, H_2_ does not leave any by-products that may affect the effluent quality [22]. Another advantage of hydrogenotrophic denitrification is that it does not require post-treatment [23] and has low sludge production. Hence, the applicability of the hydrogenotrophic denitrification process has been investigated for the removal of various pollutants such as perchlorate, selenate, chromate, arsenate, trichloroethene, trichloroethane and chloroform [24].

Lee and Rittmann [25] achieved 76% nitrate removal efficiency under 0.31 atm H_2_ pressure in synthetic wastewater containing 10 mg/L $NO_{3}^{-}–N$. When the inlet $NO_{3}^{-}–N$ concentration was increased to 12.5 mg/L and H_2_ pressure to 0.42 atm, 92% nitrate removal efficiency was obtained. Visvanathan et al. (2008) stated that hydraulic retention times of 3, 5 and 6 h, respectively, were necessary to achieve a removal efficiency of over 90% in synthetic wastewater containing 50 mg/L $NO_{3}^{-}–N$ and 1%, 2% and 3% NaCl. During these hydraulic retention times, the denitrification rates of the system were 366.8, 226.2 and 193.2 g/m^3^/day.

The stoichiometry of the hydrogenotrophic denitrification process is as follows [23]:(1)$$2NO_{3}^{-}+5H_{2}+2H^{+}\;\to \;N_{2\;\left(g\right)}+6H_{2}O$$

According to Equation (1), 1 mg of $NO_{3}^{-}–N$ requires 0.357 mg of H_2_ gas. Also, a reduction of 1 meq of $NO_{3}^{-}–N$ produces 1 meq of alkalinity (equivalent to 3.57 mg CaCO_3_).

Denitrification with hydrogen has two notable drawbacks. Firstly, hydrogen gas is not highly soluble in water. Secondly, it can be explosive in the air within the range of 4–75% [25]. The most common approach is to introduce hydrogen gas into the reactor either with a tank that is saturated with hydrogen or directly into the reactor. However, the mass transfer rate is low due to its low solubility, and there is a risk of explosion if it accumulates in a sealed place [23]. To overcome these disadvantages, membrane biofilm reactors have been developed. These reactors use gas transfer membranes to supply hydrogen, preventing the formation of bubbles. In these reactors, it has been reported that hydrogen is used with up to 100% efficiency [25].

However, membrane biofilm reactors also have some issues. In their study, Lee and Rittmann [25] found that the biofilm that developed on hollow fiber membranes was non-uniform. They found that the biofilm was thicker in the areas closest to the gas supply. The growth of biofilm on the membrane can decrease the rate of gas transfer. This can cause difficulties when there is leakage of gas from fibers or fiber regions with a thinner biofilm [22]. In this configuration, hydrogen and nitrate are supplied to opposite sides of a membrane. However, there is a challenge in that they diffuse in opposite directions, which makes the process more difficult. This means that the highest concentrations of hydrogen and nitrate are not typically found in the same place. This may limit the transfer of both substances [22]. Similarly, Visvanathan et al. [26] reported an increase of 16–31% in resistance due to fouling of the membrane used for gas transfer. However, its efficiency was restored after cleaning with chemicals.

A new MBR configuration has been developed that increases the solubility of hydrogen gas in the MBR and reduces membrane fouling [27]. In this configuration, the activated sludge suspension is circulated through a venturi injector with the headspace gas and then returned to the reactor. The cavitation in the venturi increases gas transfer, while a diffuser directs the liquid–gas mixture to the membrane surface for better scouring. However, the effect of headspace gas circulation or the means of circulation, whether by a venturi device or peristaltic pump, on membrane fouling and denitrification process performance was not studied, although MBR configuration was partly the same.

### 1.5. Venturi Injectors

A venturi tube consists of three parts: the conical narrowing region, the throat, and the conical expansion region (Figure 1). In the contraction zone, there is a decrease in pressure parallel to the increase in velocity. However, the velocity is converted back to pressure with some loss in the expansion zone.

Venturi injectors are utilized in various applications due to their ability to increase mass transfer through hydrodynamic cavitation occurring in the throat part of the device.

According to Englehart et al. [28], a typical bubble diffuser system has an O_3_ mass transfer efficiency of around 10–15%, while a system using a venturi injector can achieve a transfer efficiency of 90%. Venturi injectors have been used in full-scale wastewater treatment plants to increase oxygen transfer efficiency (e.g., [29]).

This study aimed to reduce the nitrate content of both real drainage water obtained from the Harran plain in the Southeast Anatolian Project (GAP) region and synthetically prepared drainage water through an autotrophic denitrification process with hydrogen gas. Furthermore, the impact of using a venturi-integrated MBR, a recently developed membrane bioreactor configuration, on treatment performance and membrane fouling was also investigated.

## 2. Materials and Methods

### 2.1. Feedwater Characteristics

The anoxic MBR system was fed with real agricultural drainage water (ADW) taken from an agricultural site in Harran Plain (Sanliurfa, Turkey) during the irrigation season. Table 1 shows the statistical characteristics of this real ADW, which was taken from 17 different locations and at 5 different times. Outside of the irrigation season, the MBR system was fed with synthetic agricultural drainage water (SDW). The SDW was prepared by adding NaNO_3_ to tap water to achieve a final $NO_{3}^{-}–N$ concentration of 50 or 100 mg/L.

### 2.2. Analyses

COD, $NO_{3}^{-}–N$, $PO_{4}^{3-}–P$ and SO_4_^2−^ in the samples were measured spectrophotometrically using Hach Lange test kits. The pH and conductivity of the samples were measured using a portable meter. In the MBR, pH, temperature, redox potential and dissolved oxygen were measured online. The pH was automatically adjusted to 7.5. Headspace gas samples were collected in a gas bag and measured via gas chromatography. H_2_, N_2_ and O_2_ gases were measured using an Agilent device with model no. 7890B, while CO_2_ was measured using Agilent 6890N. Membrane surface characterization was carried out via scanning electron microscope (JEOL JSM-7001F) and functional groups were identified by a FTIR spectrometer (Spectrum 100, Perkin Elmer, Waltham, MA, USA).

Hydrogen gas utilization efficiency was calculated approximately by a mass balance of H_2_ gas, assuming no accumulation of intermediate products. That is, the total mass of the H_2_ gas supplied to the reactor within a certain period should be equal to the sum of the mass of unused H_2_ gas accumulated in the gas bag and the mass of the H_2_ gas used to convert nitrate to nitrogen gas. The mass of H_2_ gas used to convert nitrate to nitrogen gas is equal to 0.357 times the mass of $NO_{3}^{-}–N$ converted to nitrogen gas, according to the general conversion stoichiometry of hydrogen-based denitrification (Equation (1)). After measuring the mass of H_2_ gas in the gas bag and calculating the mass of the H_2_ gas used for $NO_{3}^{-}–N$ converted to nitrogen gas; the ratio of the latter to their sum shows the efficiency of hydrogen gas utilization.

### 2.3. Experimental Setup

A venturi device is created by combining a converging cone with a diverging one at a location known as the throat portion. When a flow with sufficient velocity occurs in such a venturi device, the pressure at the throat portion drops below the atmospheric pressure, causing a negative gage pressure (vacuum). This vacuum draws in the fluid or gas connected to the throat portion (suction port), which is then mixed with the working fluid and exits the venturi device through the outlet.

To examine the effect of the venturi device and headspace gas circulation on MBR operation, the MBR operated at three different configurations. In two of these configurations, the headspace gas circulated in two different ways using a centrifugal pump: one via the venturi device (Figure 2a, solenoid valve open) and the other through a wye connection (Figure 2b and Figure 3). In the other configuration (Figure 2a, solenoid valve closed), the venturi device was used without headspace gas circulation. In the venturi device configurations [27], the mixed liquor of the MBR circulated through the venturi device. This configuration enabled the mixing of pure hydrogen gas, with or without headspace gas, with the mixed liquor through the suction port of the venturi device (refer to Appendix A). This highly agitated mixed liquor with pure hydrogen, with or without headspace gas, was directed to membrane surfaces in the form of jet flows through a coarse diffuser beneath the MBR. In the third configuration, where the venturi device was not used, a 30-degree wye part was mounted on the circulation line instead. In this configuration (Figure 3), headspace gas was taken in through a peristaltic pump and travelled along to the circulation line through the wye part.

The MBR system was fed with either real agricultural drainage water or synthetic water, which was delivered to the system using a peristaltic pump. The feed water temperature was kept at +4 °C by cooling. Since the amount of hydrogen gas required for the 10 L lab-scale MBR was low, a commercial device to adjust the hydrogen feed to the reactor could not be found. Instead, a solenoid valve was used to supply hydrogen gas intermittently to the reactor. Through experimentation, it was discovered that the optimal use of hydrogen occurred when the hydrogen gas supply valve was open for 2 s and closed for 25 min.

In the configuration where only pure hydrogen gas was connected to the suction port of the venturi device, the membrane surface was scoured with a liquid–gas mixture only when hydrogen gas was supplied to the system for 2 s. During the 25 min period when the hydrogen gas valve was closed, the membrane surface was scoured only with the liquid suspension. In contrast, when both pure hydrogen gas and gas accumulated in the upper part of the reactor were connected to the gas suction line of the venturi, the membrane surface was constantly scoured with a liquid–gas mixture. This approach was intended to improve the efficiency of hydrogen gas utilization and clean the membrane surface more effectively.

When using a venturi, the gas that accumulates in the upper phase of the reactor is suctioned by the venturi and delivered to the membrane surface using the diffuser system. To test whether these benefits are exclusive to the use of a venturi, the gas that accumulated in the upper phase of the reactor was drawn using a peristaltic pump instead of the venturi and introduced to the activated sludge circulation line through a 30° elbow (refer to Figure 3). This configuration was designed to compare the effects of using and not using a venturi on membrane fouling.

A flat-sheet, 0.2 μm microfiltration membrane was used in all configurations, and its properties are listed in Table 2. The MBR was operated continuously for more than 200 days, with membrane filtration being carried out using alternating 10 min on and 1 min off cycles to facilitate membrane self-cleaning. When the transmembrane pressure (TMP) reached 250 mbar, the membrane was cleaned using a NaOCl solution, followed by citric acid.

To better simulate a real MBR system, two panels of plexiglass of the same size as the membrane panel were placed on both sides of the membrane panel inside the reactor. The diffuser used had pore diameters of 3 mm, based on previous studies. Pure H_2_ gas was supplied to the system from a separate H_2_ tube, and the H_2_ gas that accumulated in the gas phase of the reactor was continuously reintroduced to the system using the venturi device.

### 2.4. Calculation of H_2_ Gas Utilization Efficiency

As the hydrogen gas requirement of the 10 L laboratory-scale anoxic denitrification system was extremely low, it was not possible to measure it directly. However, the frequency and duration of H_2_ gas introduction to the system could be adjusted using the solenoid valves. The efficiency of hydrogen utilization in the anoxic MBR system was estimated using a mass balance approach based on the amount of $NO_{3}^{-}–N$ removed and the measured gas amounts at the outlet. The details of this calculation are provided in Figure 4 and Appendix A.

## 3. Results and Discussion

### 3.1. $NO_{3}^{-}–N$ Removal Performance

As previously mentioned in Section 2.3, the MBR system operated at two different configurations when using the venturi device. In the first configuration, only hydrogen gas was connected to the suction port of the venturi injector, and gas transfer of the supplied hydrogen gas was carried out at the throat of the venturi device when the solenoid valve was open. In the second configuration, both hydrogen gas and gas that accumulated in the upper part of the reactor (headspace gas) were connected to the suction port of the venturi device. This configuration aimed to improve hydrogen gas utilization efficiency and better clean the membrane surface using a continuous liquid–gas cycle.

Figure 5 illustrates the influent and effluent $NO_{3}^{-}–N$ concentrations. When there was no headspace gas circulation, the $NO_{3}^{-}–N$ removal efficiency varied between 88 and 99%. However, when headspace gas circulation was introduced, the removal efficiency initially dropped to 19–24%. This negative effect on nitrogen removal is thought to be due to changes in hydrodynamic conditions. Specifically, the activated sludge bacteria were not exposed to hydrodynamic cavitation prior to the headspace gas circulation. However, when the system configuration changed for the first time and the gas collected at the top was drawn and circulated using the venturi device, the bacteria were subject to continuous hydrodynamic cavitation. Due to the need for adaptation to these changing conditions, it is estimated that there was a decrease in nitrogen removal efficiency. As seen in Figure 5, the effluent nitrate concentrations began to decrease again from the 25th day of operation. When Figure 5 and Figure 6 are evaluated together, it is evident that there was no negative effect on nitrate removal efficiency in the absence of headspace gas circulation, even when the dissolved oxygen concentration ranged between 0.75 and 0.90 mg/L. However, with headspace gas circulation, the oxygen concentration reached 1.6 mg/L (Figure 6). The increase in oxygen concentration in the case of the headspace gas circulation is due to the entry of oxygen into the reactor from the outside when the vacuum is applied to the gas phase of the reactor. This can occur even through a very small pore due to the pressure difference that has formed unless a very good seal is provided. As depicted in Figure 6, the average oxygen concentration was measured as 0.82 mg/L on the 38th day. In ongoing studies, improvements in the reactor insulation were made in parallel with efforts to improve nitrate removal. These improvements led to the attainment of high nitrate removal efficiencies, even in cases where gas recirculation was present. This was likely due to both the adaptation of the microorganisms to the changing conditions caused by the gas recirculation and the reduction in the oxygen concentrations in the reactor to levels of 0.2–0.4 mg/L, which provided a more stable environment.

As seen in Figure 5, during the first period (days 17–56), when the headspace gas circulation was applied, the effluent $NO_{3}^{-}–N$ concentration dropped below 10 mg/L from the 45th day, and the $NO_{3}^{-}–N$ removal efficiency increased up to 98%. When the inflow $NO_{3}^{-}–N$ concentration was increased to 100 mg/L, the effluent $NO_{3}^{-}–N$ concentration initially rose to 31.5 mg/L, but then fell to 7.9 mg/L (Figure 5). The maximum removal efficiency of 92% was achieved at a $NO_{3}^{-}–N$ concentration of 100 mg/L.

Figure 7 illustrates the changes in ORP throughout the study. ORP and dissolved oxygen concentration appear to follow a similar trend. That is to say, if other conditions remain relatively constant, it has been observed that ORP tends to become positive when oxygen levels increase, and negative when oxygen levels decrease. Negative ORP values between 250 and 500 mV were observed when nitrate removal was well achieved.

### 3.2. The Effect of Headspace Gas Circulation on Membrane Fouling

Figure 8 shows the effect of headspace gas recirculation on the change in transmembrane pressure (TMP). In the absence of headspace gas circulation in an anoxic MBR, the membrane surface was only cleaned with liquid flow. However, chemical cleaning was required after an average of 5 days, since more than a 20% decrease in membrane flux was observed. In contrast, when the headspace gas circulation was present, the gas accumulated in the upper part of the reactor was vacuumed by the venturi and the resulting bubbled liquid–gas stream was used to continuously scour the membrane surface. It was observed that the TMP of the membrane, which was chemically cleaned on average every 5 days before, did not increase for 5 days, and the need for chemical washing occurred after 18 days on average.

The effect of scouring the membrane surface with gas on membrane fouling has been investigated in the literature. Vinardell et al. [16] studied the effect of specific nitrogen gas flow rates varying between 0.25 and 2.0 m^3^/m^2^·h on membrane fouling at membrane fluxes ranging from 5 to 20 L/m^2^·h in a granular anaerobic membrane bioreactor. Their study found that membrane fouling depended on both the membrane flux and the specific gas flow rate. The most economical operating conditions were found at a medium flux (7.8 L/m^2^·h) and specific gas flow rates (0.5 m^3^/m^2^·h). Similarly, Wang et al. [17] investigated the effect of intermittent stripping of the membrane surface with nitrogen gas (providing gas only in the case of membrane relaxation) in a granular anaerobic membrane bioreactor on membrane fouling compared to continuous gas use. They found that intermittent stripping of the membrane surface (10 s on/10 s off) reduced energy consumption by 50%, while providing sustainable fluxes close to continuous gas usage. Another study by Thongsai et al. [30] investigated the effects of intermittently circulating the reactor fluid for various durations (15–45 min or continuously on, 15 min off) on membrane fouling, while keeping the specific gas flow rate constant (0.3 m^3^/m^2^·h), in an anaerobic membrane bioreactor. They found that the least membrane fouling occurred when the liquid circulation was closed for 15 min. These studies suggest that the timing and duration of gas or liquid scouring can play a critical role in reducing membrane fouling and improving system performance.

### 3.3. The Effect of Venturi Usage on Membrane Fouling

When the headspace gas was taken with a peristaltic pump instead of a venturi, given to the activated sludge circulation line and recycled, the membrane fouled more rapidly (Figure 9). Figure 8 shows that during the period of synthetic drainage water, chemical membrane cleaning was performed every 12 days on average, including technical problems. However, when the venturi device was not used, it was observed that chemical cleaning of the membrane was required every 1–2 days, even if the headspace gas was recycled. This result highlights the importance of ensuring a sufficient velocity of liquid–gas flow used in scouring the membrane surface to limit membrane fouling. The use of a venturi device appears to be an effective way to achieve this goal.

SEM images of the membranes and EDS graphics reveal that inorganic pollutants accumulate on the surface of the used membrane (Figure 10 and Figure 11). This observation is clearly supported by EDS charts. After the chemical cleaning process was applied to the used membrane, it was observed that the accumulated foulants on the surface were successfully removed. These findings suggest that chemical cleaning is an effective way to remove inorganic pollutants and other foulants from the membrane surface, thus improving system performance.

The FT-IR spectra (Appendix A) suggest that the membranes can be successfully returned to their original structure and reused after the chemical cleaning process was applied to them. The foulants observed on the membrane surface do not appear to affect the chemical structure of the membrane. Therefore, it is possible to return the membrane to its original state by chemical washing, ensuring that the membrane can be reused and improving the overall sustainability of the system.

As stated in Section 2.2, it was not possible to find a commercial gas flowmeter to measure the low flow rate of the hydrogen gas supplied to the reactor. However, headspace gas analysis was performed to indirectly measure the amount of hydrogen used in the reactor. At the beginning of the operation, hydrogen gas was continuously supplied to the reactor under 30 mbar pressure. However, continuous hydrogen supply resulted in an inefficient use of hydrogen, since the inlet nitrate concentration was low compared to the amount of gas supplied. Based on these results, it was decided to supply the hydrogen gas to the reactor intermittently. Inlet and outlet valves were installed, which made it possible to set how long hydrogen would be supplied to the system and how long the gas would be kept in the system. By doing so, it was possible to optimize hydrogen usage and improve system efficiency.

### 3.4. The Effect of Venturi Usage on Hydrogen Gas Utilization Efficiency

In the first trial using the venturi, hydrogen was supplied to the system under 100 mbar pressure for 4 s, and this gas was kept in the system for 5 min. Analysis of the headspace gas showed 95.9% of the total gas was H_2_ gas (Table 3). The high percentage of hydrogen gas in the output gas indicated that the hydrogen gas supplied to the system was quite high.

The high rate of hydrogen output in the outlet gas was likely due to a large amount of gas being supplied to the reactor, which indicated that hydrogen gas was not a limiting factor. This conclusion is supported by the treatment efficiency achieved in the system, as shown in Figure 5. For example, on the 170th day, the inlet $NO_{3}^{-}–N$ concentration was 94.5 mg/L, while the output concentration was 2.6 mg/L, indicating a very high treatment efficiency (97%). However, as shown in Table 3, the amount of H_2_ gas in the outlet gas decreased when the gas holding time was increased to 25 min, indicating that the amount of hydrogen gas supplied to the system was optimized. After some trials were carried out to maximize the hydrogen gas utilization efficiency in an anoxic MBR, the hydrogen gas utilization efficiency was increased to 96%. This yield value was calculated using the results of a gas analysis, where the partial pressures of hydrogen and nitrogen gases were 17 and 76.6%, respectively (Table 3). The influent $NO_{3}^{-}–N$, effluent $NO_{3}^{-}–N$ and effluent *NO*_2_–*N* concentrations were 102, 9.3 and 2.3 mg/L, respectively, and the calculation method is given in Section 2.4. Overall, optimizing the supply and utilization of hydrogen gas can improve system efficiency and treatment performance.

## 4. Conclusions

This study investigated the use of hydrogen gas in a submerged membrane bioreactor with an integrated venturi for nitrate removal from agricultural drainage waters. In both real and synthetic drainage water studies, with an average influent $NO_{3}^{-}–N$ concentration of around 50 mg/L, 88–99% removal efficiency was achieved in the absence of headspace gas circulation. When headspace gas circulation was used, the removal efficiency increased up to 98% after the acclimation period. When the influent $NO_{3}^{-}–N$ concentration was increased to 100 mg/L, a maximum removal efficiency of 92% was achieved, even when the headspace gas was circulated.

The bioreactor configuration used in this study not only increases the solubility of hydrogen gas in water, but also helps reduce membrane fouling by providing better scouring of the membrane surface with the jet stream it creates. This dual advantage, provided simultaneously, may contribute to the widespread use of anoxic/anaerobic submerged membrane bioreactors in water/wastewater treatment.

In future studies, it is recommended to use better-isolated reactors (possibly using reactor parts produced by a CNC machine), in which the dissolved oxygen concentrations in the reactor do not increase when a vacuum is applied to achieve better treatment performance.

## Figures and Tables

**Figure 1 membranes-13-00666-f001:**
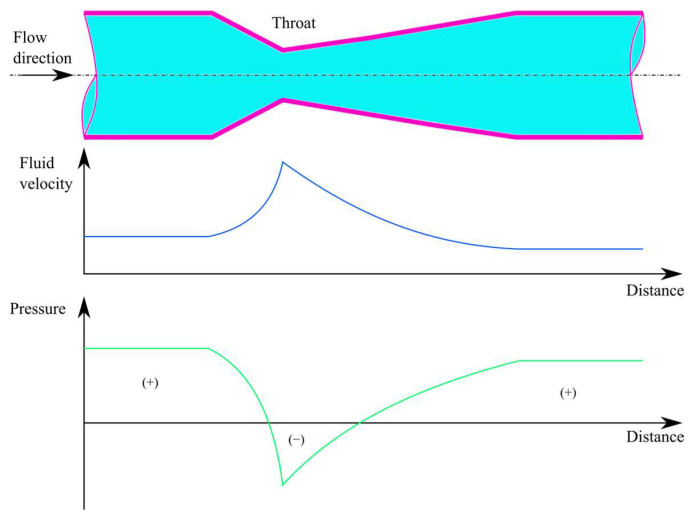
The change in fluid velocity and pressure along the flow direction in a venturi device.

**Figure 2 membranes-13-00666-f002:**
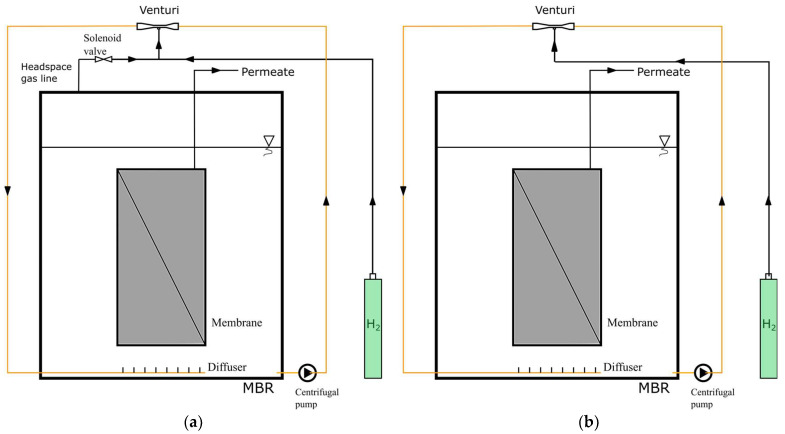
MBR configurations: (**a**) venturi device with headspace gas circulation (solenoid valve open), venturi device without headspace gas circulation (solenoid valve closed), and (**b**) headspace gas circulation without the venturi device.

**Figure 3 membranes-13-00666-f003:**
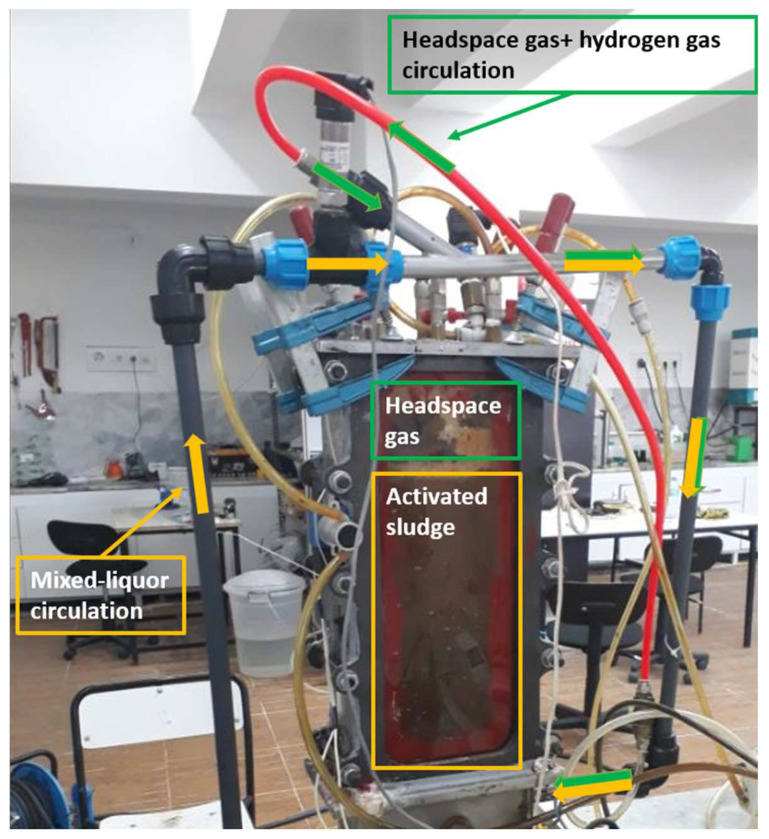
Headspace gas circulation via a peristaltic pump.

**Figure 4 membranes-13-00666-f004:**
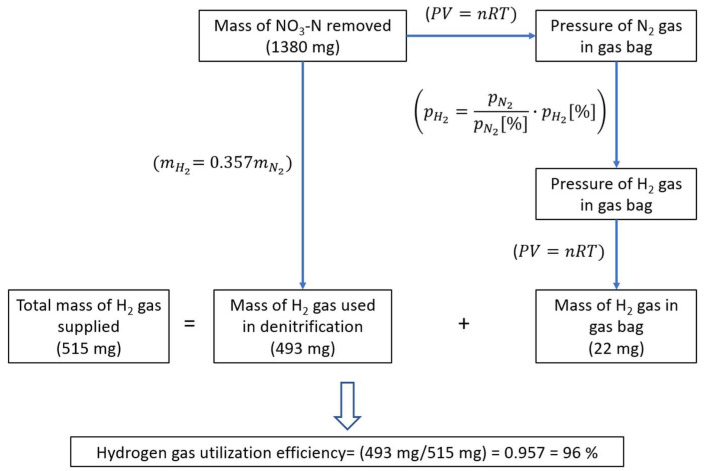
Approximate calculation of H_2_ gas utilization efficiency by establishing a mass balance based on the amount of denitrified nitrogen and the headspace gas measurement.

**Figure 5 membranes-13-00666-f005:**
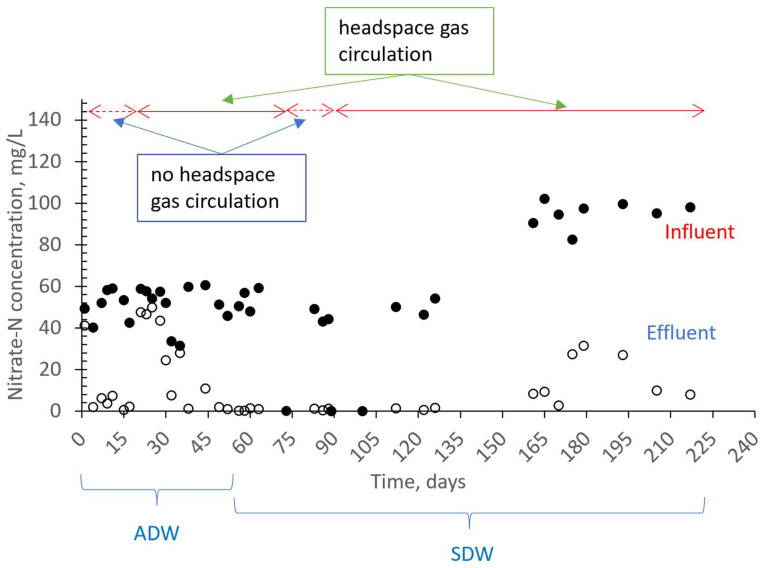
Effect of headspace gas circulation on $NO_{3}^{-}–N$ removal from real/synthetic agricultural drainage water.

**Figure 6 membranes-13-00666-f006:**
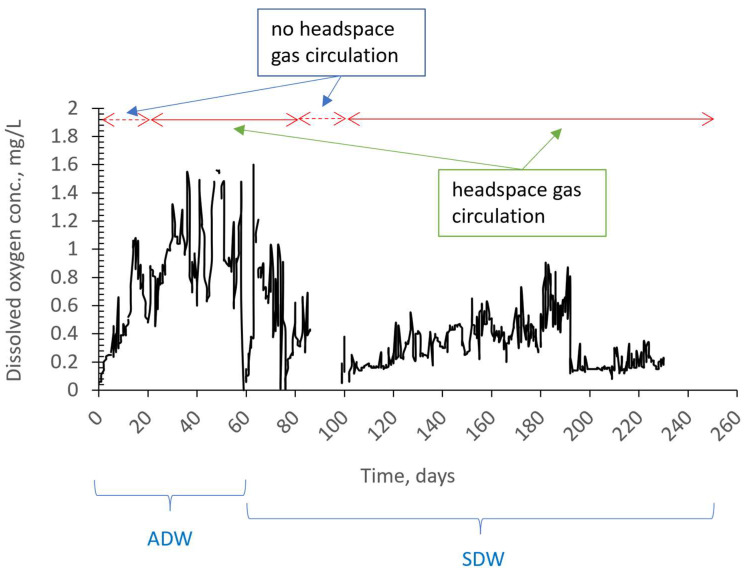
Dissolved oxygen concentration in an anoxic MBR during operation with/without headspace gas circulation.

**Figure 7 membranes-13-00666-f007:**
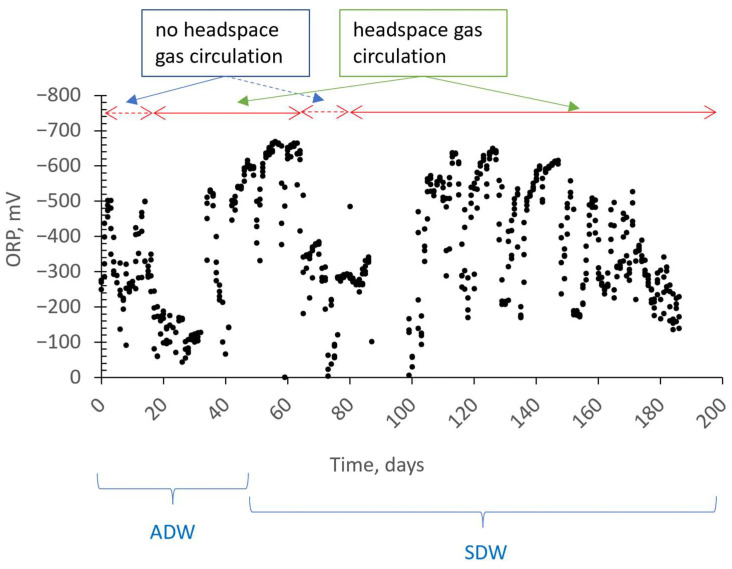
Oxidation–reduction potential in an anoxic MBR during operation with/without headspace gas circulation.

**Figure 8 membranes-13-00666-f008:**
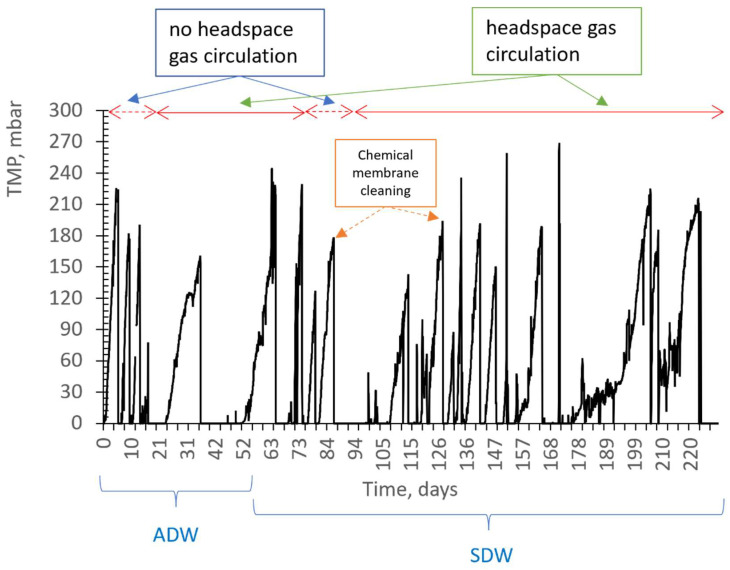
TMP change venturi device configurations with and without headspace gas circulation.

**Figure 9 membranes-13-00666-f009:**
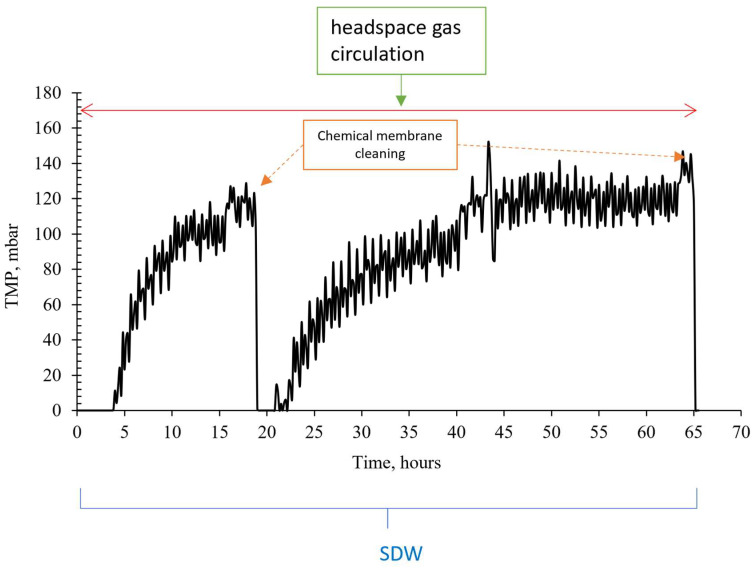
Variation of transmembrane pressure in the case of headspace gas circulation with a peristaltic pump instead of a venturi device.

**Figure 10 membranes-13-00666-f010:**
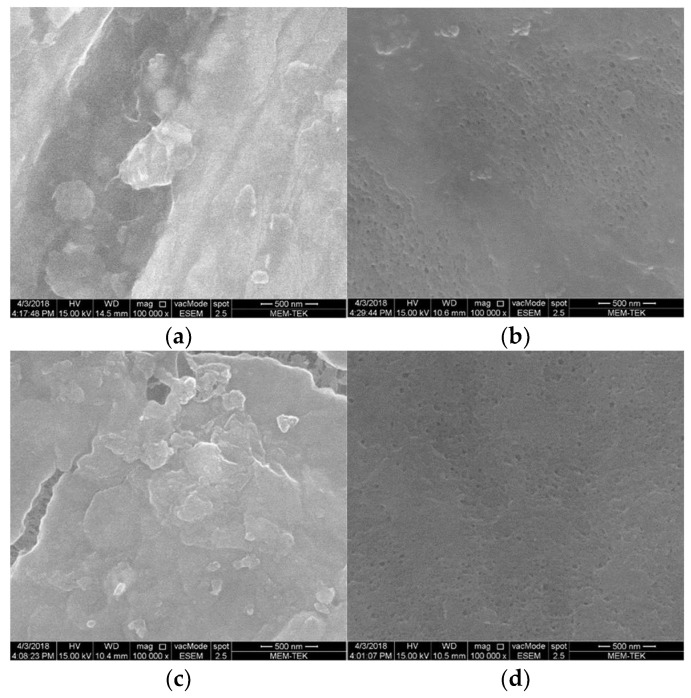
SEM images of the fouled (**a**) and cleaned membrane (**b**) in the absence of headspace gas circulation using the venturi device, and fouled (**c**) and cleaned (**d**) membrane in case of headspace gas circulation using the venturi device (magnification: 100,000×). (Original images are provided on Appendix A).

**Figure 11 membranes-13-00666-f011:**
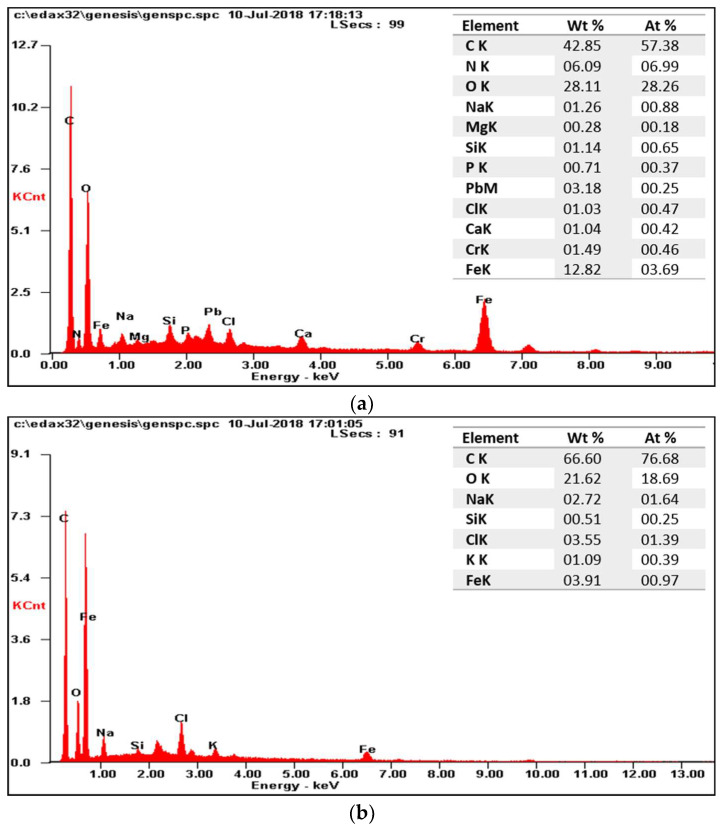
EDS graphs of a (**a**) fouled and cleaned (**b**) membrane.

**Table 1 membranes-13-00666-t001:** Characteristics of real agricultural drainage water.

Parameter/Statistic	pH	Conductivity (μS/cm)	COD (mg/L)	$NO_{3}^{-}–N$ (mg/L)	$PO_{4}^{3-}–P$ (mg/L)	SO_4_^2−^ (mg/L)
Average	7.7	2872	10.9	13.1	0.3	652
Minimum	7.1	654	2.5	2.6	0.0	80
Maximum	8.4	7130	47.8	44.7	1.4	1200
Standard deviation	0.3	1750	7.6	9.7	0.4	301

**Table 2 membranes-13-00666-t002:** Properties of the membrane used in the MBR.

Property	Value
Membrane Element	SINAP–10–PVDF
Effective surface area, m^2^	0.1
Dimensions, W × L × D, mm	220 × 320 × 6
Weight, kg	0.4
Pore size, µm	0.1
Capacity, L/min/pc	40–60
Aeration volume, L/min/pc	6
pH	3–12
Effluent turbidity, NTU	1.0
Effluent TDS, mg/L	1

**Table 3 membranes-13-00666-t003:** Headspace gas analysis in an anoxic MBR.

H_2_ Gas Supply Valve Open/Closed	Gaz Component (%)					
	H_2_	O_2_	N_2_	CO_2_	Others ^†^	Total
4 s/5 min	95.9	0.97	3.13	-	-	100
2 s/25 min	17.0	3.4	76.6	1.7	1.3	100

^†^: Sulfurous compounds, e.g., were not measured.

## Data Availability

The author confirms that the data supporting the findings of this study are available within the article and/or its Appendix A.

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
