# Peer review of "Anoxic Treatment of Agricultural Drainage Water in a Venturi-Integrated Membrane Bioreactor"

_membranes, 2023, doi:10.3390/membranes13070666_

Round 1

Reviewer 1 Report

The paper is relatively well written and the idea of using a venturi injector is interesting. However, there are several issues to be addressed: 

1. the abstract and the introduction is lengthy. the introduction on venturi device may not be necessary as it is basic engineering knowledge. 

2. line 370, the authors mentioned the leakage of the reactor, and later fixed the problem. Why not start over the experiment with the sealed reactor again, instead of reporting the possibly flawed data?

3. line 390, 'Negative ORP values 390 between 430 and 450 mV were observed when nitrate removal was well achieved', Figure 7 shows large variations, where do the authors isolate such a small gap on the figure? 

4. Figure 9, please explain the data where spikes can be observed, e.g. around day 170? and the slow increase in TMP after that. 

5. line 449, have the authors measured the velocity of liquid and gas flow of the two different setup? Please provide the comparison. 

6. line 492, only the data on 170th day was mentioned while the data in other days were not as good. One can see significant fluctuation of data, the optimal value is not representative. 

7. overall, the reactor design is worth studying, while the advantage of the venturi device is not convincingpy or comprehensively demonstrated. More detailed works may be carried out in the future work.  

Reviewer 2 Report

TITLE -  refers to the content of the manuscript.

INTRODUCTION - is  a proper introduction to the topic of work; it does   specify the subject matter; is too long and should be shortened.

MATERIALS AND METHODS - presented at a good level; there is no information on how many attempts were made to present the characteristics of real agricultural drainage water.

RESULTS AND DISCUSSION - are satisfactory; the description of figures is clear and understandable.

CONCLUSIONS -  are general; the innovativeness of the conducted research was not explained; the conclusion section lacks a perspective for future research; can be improved.

REFERENCES - are adequate, correct and in line with the previous chapters. The use of the bibliography is correct.

Recommendation

I recommend this manuscript for publication in the journal Membranes with minor corrections.

Reviewer 3 Report

This MS tried to investigate the nitrogen removal  from anoxic MBR. The idea is ok, but should be carefully revised before it can be reconsidered.  

1. The abstract is too long and general. It should focus the significance  and important achievement of this study.

2. Introduction section should be shorten.

3. NO3-N, PO4-P should be  NO3--N, PO43--P.

4. In Fig.6, OD was detected, but why ORP is low in Fig. 7?

5. Fig. 8 is not related to nitrogen removal and shoudl be removed.

6. Fig. 11 is not obvious to reflect the results

7. More discussion on the relationships between Fig. 12 and membrane fouling should be added.

8. Conclusions section is too long.

9. The manuscript should be carefully revised not only the structure but also the writting.

10. The quality of the Figures should be improved.

language should be improved
